# The Impact of microRNAs on Mitochondrial Function and Immunity: Relevance to Parkinson’s Disease

**DOI:** 10.3390/biomedicines11051349

**Published:** 2023-05-03

**Authors:** Beatriz F. S. Guedes, Sandra Morais Cardoso, Ana Raquel Esteves

**Affiliations:** 1CNC–Center for Neuroscience and Cell Biology and CIBB-Center for Innovative Biomedicine and Biotechnology, University of Coimbra, 3004-504 Coimbra, Portugal; 2Institute of Cellular and Molecular Biology, Faculty of Medicine, University of Coimbra, 3004-504 Coimbra, Portugal; 3IIIUC–Institute for Interdisciplinary Research, University of Coimbra, 3004-504 Coimbra, Portugal

**Keywords:** Parkinson’s disease, gut–brain axis, mitochondria, immunity, microRNAs, gut microbiota

## Abstract

Parkinson’s Disease (PD), the second most common neurodegenerative disorder, is characterised by the severe loss of dopaminergic neurons in the Substantia Nigra pars compacta (SNpc) and by the presence of Lewy bodies. PD is diagnosed upon the onset of motor symptoms, such as bradykinesia, resting tremor, rigidity, and postural instability. It is currently accepted that motor symptoms are preceded by non-motor features, such as gastrointestinal dysfunction. In fact, it has been proposed that PD might start in the gut and spread to the central nervous system. Growing evidence reports that the gut microbiota, which has been found to be altered in PD patients, influences the function of the central and enteric nervous systems. Altered expression of microRNAs (miRNAs) in PD patients has also been reported, many of which regulate key pathological mechanisms involved in PD pathogenesis, such as mitochondrial dysfunction and immunity. It remains unknown how gut microbiota regulates brain function; however, miRNAs have been highlighted as important players. Remarkably, numerous studies have depicted the ability of miRNAs to modulate and be regulated by the host’s gut microbiota. In this review, we summarize the experimental and clinical studies implicating mitochondrial dysfunction and immunity in PD. Moreover, we gather recent data on miRNA involvement in these two processes. Ultimately, we discuss the reciprocal crosstalk between gut microbiota and miRNAs. Studying the bidirectional interaction of gut microbiome–miRNA might elucidate the aetiology and pathogenesis of gut-first PD, which could lead to the application of miRNAs as potential biomarkers or therapeutical targets for PD.

## 1. Introduction

### Parkinson’s Disease

Parkinson’s disease (PD) was first described by James Parkinson more than two centuries ago, in “An Essay on the Shaking Palsy” in 1817. PD is the second most common progressive neurodegenerative disorder, after Alzheimer’s Disease, with more than 10 million people affected worldwide in 2021. Its incidence and prevalence increase steadily with age (median age at onset is 60 years old), and men are 1.5 times more likely to suffer from the disease than women [1]. 

PD is clinically characterised by the development of numerous motor symptoms, including bradykinesia, resting tremor, rigidity, and postural instability [2]. These motor symptoms develop as a consequence of the severe loss of dopaminergic neurons in the Substantia Nigra *pars compacta* (SNpc), and only emerge when striatal dopamine levels are decreased by 60–70% as a result of the degeneration of 40–60% of neurons in the SNpc [3]. The presence of intracytoplasmic protein inclusions of α-synuclein, known as Lewy bodies (LBs), is another important pathological hallmark of PD. Furthermore, it is believed that PD progression might also affect non-dopaminergic pathways prior to the onset of nigral neurodegeneration, leading to the manifestation of several non-motor symptoms that appear at a prodromic stage, up to 15–20 years before the onset of motor symptoms (Figure 1) [2]. Based on various population studies, a multitude of non-motor symptoms have been commonly associated with prodromal PD, namely, hyposmia, constipation, depression, dysautonomia, and rapid eye movement (REM) sleep behaviour disorder (RBD) (Figure 1) [2,3].

Since the discovery of LBs, it has been clear that a variety of other neuronal populations were affected by these proteinaceous inclusions, and not only the neurons in the SNpc [4]. In order to determine whether the pathology affects the nigral and extranigral structures simultaneously, Braak and colleagues conducted various studies, which culminated in the formulation of a six-stage hypothesis for the progression of Lewy pathology (LP) in sporadic PD [5]. According to Braak’s hypothesis, PD pathogenesis might initiate in non-dopaminergic areas, such as the olfactory bulb and the enteric nervous system (ENS), years and even decades before spreading, via the olfactory tract and the vagal nerve, respectively, towards the central nervous system (CNS), eventually reaching the SNpc and triggering the emergence of motor symptoms [6]. The earliest stages of LP (before SNpc involvement) have been linked to the prodromal non-motor symptoms of PD, such as olfactory impairment and gastrointestinal dysfunctions [7], since consistent evidence has demonstrated the presence of α-synuclein aggregates in the gut and in the neurons of the olfactory bulb during the prodromal phase, before the clinical diagnosis of PD [6,8]. Nonetheless, studies showed that some PD cases cannot be staged by the Braak staging system, as some patients presented with evident LB pathology first in the locus coeruleus, SNpc, and the amygdala, and then in the dorsal motor nucleus of the vagus (DMV) and gut, representing a rostro-caudal route of α-synuclein transmission from the CNS to the ENS [9]. This neuropathological evidence gave rise to two distinct hypotheses: body-first PD vs. brain-first PD. In the body-first subtype of PD, LP begins in the enteric or autonomic nervous system and spreads to the CNS via the vagal nerve, which aligns with Braak’s proposed staging [10]. These patients display more autonomic symptoms and a longer prodromal phase, with the development of non-motor symptoms such as gastrointestinal dysfunction, RBD, and hyposmia [9]. The opposite is true for the brain-first subtype of PD, where brain pathology presumably initiates in the amygdala or in closely connected areas such as the olfactory bulb [9]. These patients present with fewer autonomic symptoms and a shorter prodromal phase, with less frequent development of non-motor symptoms such as RBD or hyposmia [9]. Present evidence reveals that most PD cases follow the body-first subtype [10].

Despite all the efforts and contributions of the scientific community, the primary cause of sporadic PD remains largely unknown; however, it has been hypothesised that the disease has a multifactorial aetiology, resulting from the interaction between environmental, genetic, and age-associated factors [8]. Several environmental and lifestyle risk factors have been linked to PD, including environmental toxins, pesticide and heavy metals exposure, traumatic lesions, and bacterial or viral infections [11]. An interesting report from 2022 evaluated the correlation between some risk factors, including coffee consumption, cigarette smoking, and physical activity, and clinical symptoms [12]. Remarkably, coffee consumption is positively correlated with milder motor symptoms, whereas physical exercise is positively correlated with milder non-motor symptoms. 

Moreover, while most cases occur in a sporadic manner, approximately 5–10% of PD cases are of genetic origin [11]. The era of PD genetics started when several families seemed to exhibit a Mendelian inheritance pattern (dominant or recessive) of Parkinson’s, suggesting a genetic cause of the disease [2]. In 1997, the first PD-associated mutation was discovered in the *SNCA* gene [13], originating a highly productive period of gene hunting that resulted in the identification of several PD-related genes, with either autosomal dominant (e.g., *SNCA*, *LRRK2*, *VPS35*) or autosomal recessive (e.g., *PRKN*, *PINK1*, *DJ1*) modes of inheritance [14]. Although genetic PD corresponds to a small fraction of all cases, focusing the research on genetic forms has proved to be valuable to understanding the pathophysiology of PD, since some of the proteins encoded by PD-associated genes are involved in key neuropathological mechanisms linked to the development and progression of both sporadic and familial PD [15]; these include α-synuclein aggregation and accumulation, mitochondrial dysfunction, proteasomal and autophagic impairment, oxidative stress, and neuroinflammation [2]. 

For several decades, numerous epidemiological studies have consistently shown that increased PD prevalence positively correlates with increasing age, highlighting aging as one of the most important contributing factors for PD. Indeed, the aging process is strongly associated with mitochondrial dysfunction, increased oxidative stress, neuroinflammation, and impairment of protein clearance (which facilitates α-synuclein accumulation even in the normal ageing brain [16]), all of which have also been considered pathological hallmarks of PD [4,17].

James Parkinson optimistically stated that “there appears to be sufficient reason for hoping that some remedial process may ere long be discovered, by which, at least, the progress of the disease may be stopped” [18]. Over 200 years later, no cure or neuroprotective therapy has been discovered for PD; nevertheless, researchers have been making huge progress in understanding neurodegeneration in PD, which will bring us closer to finding new effective disease-modifying therapies [8]. The currently available diagnostic methods are not able to predict the onset of the disease early on; therefore, the existing pharmacological and neurosurgical treatments are symptomatic, in order to ameliorate motor and non-motor symptoms and attempt to slow down the progression of the neurodegenerative process, but are not able to fully stop it [1].

The relationship between microRNAs (miRNAs) and PD has been intensively studied for several years, leading to the hypothesis that miRNAs might play an important role in the pathogenesis and progression of PD. Numerous miRNAs found to be dysregulated in PD were shown to target key molecular mechanisms linked to the pathogenesis and progression of PD, namely, α-synuclein aggregation and accumulation, mitochondrial dysfunction, oxidative stress, and inflammation. Herein, we will summarize evidence regarding mitochondrial dysfunction and inflammation involvement in PD and, in addition, will review the impact of miRNA dysregulation in these two important molecular mechanisms implicated in PD pathophysiology.

To carry out this state-of-the-art review, which aims for a comprehensive search of the current literature, we performed a methodical search of the literature using the database PubMed. Relevant articles were searched without year limitations. The search strategy consisted of a combination of the keywords “Parkinson’s disease” together with (AND) “mitochondrial dysfunction” (OR) “inflammation” (OR) “innate and adaptive immunity”; “Parkinson’s disease” together with (AND) “miRNA” (AND) “mitochondrial dysfunction” (OR) “inflammation” (OR) “innate and adaptive immunity” (OR) and (AND) “gut microbiota”. The present review includes an extensive collection and evaluation of the published studies (1245 papers), where 210 were used, and evaluations were independently performed by two authors (Guedes BFS and Esteves AR).

## 2. Mitochondrial Dysfunction in Parkinson’s Disease

Mitochondria are dynamic membrane-bound organelles known to play a critical role in multiple cellular functions, including ATP production, Ca^2+^ homeostasis, ROS generation, and programmed cell death [19]. The first evidence that mitochondrial dysfunction is a key player in PD pathogenesis dates back to 1983, when Langston and colleagues [20] reported several cases of a parkinsonian syndrome in humans caused by the use of a synthetic drug containing 1-methyl-4-phenyl-1,2,3,6-tetrahydropyridine (MPTP). These events rapidly received the attention of the scientific community, leading to a better understanding of how MPTP caused parkinsonism. The lipophilic nature of MPTP allows it to cross the blood–brain barrier and, once inside the brain, it is metabolised to 1-methyl-4-phenylpyridinium (MPP^+^), which is taken up by dopaminergic neurons through the dopamine transporter (DAT), inhibiting complex I activity from the mitochondrial electron transport chain (ETC) and resulting in energy depletion and oxidative stress [19]. Mitochondrial dysfunction associated with PD pathogenesis can result from ETC dysfunction, impaired mitochondrial biogenesis, increased oxidative stress, defective mitophagy, altered mitochondrial dynamics, compromised trafficking, and aberrant Ca^2+^ homeostasis [21]. The complex interplay between these compromised cellular functions leads to progressive cellular dysfunction that ultimately results in dopaminergic neurodegeneration that underlies PD pathogenesis.

The hypothesis that PD can be triggered by mitochondrial dysfunction [22] was validated when decreased activity of complex I was found in PD brain samples [23,24,25,26], platelets [27,28,29,30], lymphocytes [30], fibroblasts [31,32], skeletal muscle [33,34,35], and PD cytoplasmic hybrids (cybrids) [36]. Cybrids are created through transfer of platelet mitochondria from either PD or control subjects to mitochondrial DNA (mtDNA)-depleted recipient cells (rho0 cells). The end result are cybrid cell lines that express the nuclear genes of the recipient rho0 cell line and the mitochondrial genes of the platelet donor [37]. In addition to MPP^+^, other toxins and pesticides (like rotenone and paraquat) that impair mitochondrial complex I activity also cause parkinsonism symptomatology and dopaminergic neuron loss in animals [38]. Mitochondrial dysfunction may be potentiated by impaired mitochondrial biogenesis, a complex process essential for the maintenance of a healthy mitochondrial network [19]. Mitochondrial respiration, mitochondrial antioxidant defence, as well as mitochondrial biogenesis are regulated by PGC-1α (peroxisome proliferator-activated receptor-gamma coactivator-1alpha), which is significantly reduced in diverse PD models [39,40,41,42], therefore adversely affecting cellular bioenergetics. On the other hand, PGC-1α overexpression has been shown to alleviate α-synuclein oligomerization [39] and protect dopaminergic neurons [43].

The production of ROS through mitochondrial respiration is physiologic; however, disruption of respiratory chain complexes causes excessive production of ROS and leads to oxidative stress, which is detrimental to cells [44]. Remarkably, mitochondria are, simultaneously, the main source and the primary targets of ROS. Under normal conditions, antioxidant proteins like superoxide dismutase (SOD) and glutathione (GSH) prevent ROS levels from rising too high, but malfunction of these defence mechanisms leads to oxidative stress. Consequently, high ROS levels damage the components of the respiratory chain, as well as mtDNA. Complex I was found to be oxidatively damaged in PD brain biopsies, which compromises its appropriate assembly and function [25], leading to further inhibition and greater ROS production. In addition, post-mortem samples from PD patients displayed increased lipid peroxidation with glutathione pathway impairment [45]. ROS possess the ability to damage mtDNA by causing single- and double-strand breaks that, when repaired inefficiently, originate mtDNA deletion mutations that affect the genes coding for essential proteins involved in the mitochondrial respiratory chain [44]. The mitochondrial genome is packaged in nucleoids by the mitochondrial transcription factor A (TFAM), in order to protect it from oxidative insults; however, TFAM deficiency has been observed in dopaminergic neurons of sporadic PD subjects [46,47], suggesting an enhanced exposure of mtDNA to ROS damage in PD. Several studies have detected accumulation of deletions in the mtDNA of aged SNpc neurons of both healthy and PD individuals [46,48,49,50]. Additionally, mtDNA copy numbers have been found to increase with age in healthy individuals [50], allowing the neurons to adapt to mitochondrial dysfunction and maintain a healthy population of mitochondria. However, failure of such regulatory mechanisms of mtDNA copy numbers has been found in PD patients, leading to mtDNA copy number depletion [46,48,50,51], despite accumulating mutations. Evidence has shown that oxidative stress is linked to dopamine metabolism, justifying the selective neurodegeneration of dopaminergic neurons [52]. Auto-oxidation of dopamine generates free radicals and active quinones, which in turn interact with ROS scavengers, respiratory chain complexes, and proteins involved in mitophagy [52], contributing to increased levels of ROS and defective mitophagy processes. Indeed, autophagic alterations occur in PD, as demonstrated by an increased number of autophagosomes observed in post-mortem PD patient brains [53]. Moreover, substantia nigra neurons from PD patients display abnormal mitochondria within autophagosomes, indicating defects in mitophagy [54]. Mitophagy alterations were also observed in several PD models, including toxin-induced and genetic models [55,56]. Most interestingly, PD-associated gene mutations such as PINK1 and Parkin are known to be involved in this selective process of mitochondrial degradation [57]. Remarkably, α-synuclein is also a known target of autophagic degradation [58]. 

Oxidative damage has also been implicated in impaired proteasomal ubiquitination and degradation of proteins, facilitating the accumulation of aggregated α-synuclein in the form of LBs [59]. α-synuclein accumulation has been noted to induce redox imbalance and mitochondrial fragmentation in in vitro and in vivo models of PD [60,61]. Moreover, α-synuclein has been proven to build up inside mitochondria, damaging complex I of the ETC, resulting in mitochondrial dysfunction and oxidative stress [62,63,64].

Oxidative stress unquestionably contributes to PD pathology, and it is now acknowledged that ROS are a result of mitochondrial dysfunction and further contribute to exacerbating cell death [65]. The interaction between these numerous mechanisms associated with mitochondrial impairment forms a positive feedback loop that drives cellular dysfunction, resulting in the development and progression of PD. 

Importantly, as previously mentioned, men have a higher risk of developing PD than women, and mitochondria seem to play a significant role in this fact. Interestingly, brain mitochondria from female mice and post-mortem samples have higher electron transport chain activity and lower oxidative stress [66]. A recent transcriptomic study observed that male PD patients show increases in oxidative stress, inflammation, and angiogenesis when compared to females [67].

## 3. Innate and Adaptive Immunity Activation in Parkinson’s Disease

The immune system is the result of the complex interplay between innate and adaptive immunity. The innate immune system, which is the first line of defence, is ancient, highly conserved, and nonspecific. It consists of tissue-resident macrophages, dendritic cells, monocytes, granulocytes, and neutrophils. Interestingly, in addition to immune cells, there is a plethora of non-immune cells capable of inducing innate immune responses, such as epithelial cells, epidermal keratinocytes, mesenchymal cells, stromal cells, and neurons [68]. A significant amount of evidence has demonstrated the expression of pattern recognition receptors (PRRs), such as Toll-like receptors (TLRs), on intestinal epithelial cells and neurons, suggesting that these non-immune cells possess adequate machinery to act against pathogens through the activation of innate immunity checkpoints [69]. Remarkably, the expression of TLR4 has been found to be altered in inflammatory bowel diseases patients [70]. Colonic epithelial cells isolated from these patients were shown to secrete Interleukin (IL)-1β, produced through activation of the inflammasome [69]. Likewise, it has been demonstrated that neurons express TLR3 and TLR4 [71,72] and also produce inflammatory cytokines, such as IL-6, tumour necrosis factor alpha (TNF-α), and interferon, that may mediate innate immunity in the absence of microglial cells [73]. Furthermore, evidence shows that neurons can activate NF-κB-dependent nucleotide-binding oligomerization domain leucine-rich repeat and pyrin domain-containing protein 3 (NLRP3) inflammasome under ischemic conditions without microglia contribution [74]. Moreover, work from our group demonstrated that both enriched mesencephalic and cortical neuronal cultures exposed to BMAA, a bacterial neurotoxin, activated the NLRP3 inflammasome, accompanied by the release of IL-1β, indicating that neurons can initiate this activation without microglia contribution [71,72]. 

Overall, the innate immune system provides a rapid line of defence against threats to the host, while adaptive immunity is highly specific and is able to recognize and memorize a specific pathogen and effectively produce an immune response against the previously encountered pathogen [75]. The innate and adaptive immune systems work very closely to produce effective and protective immune responses. However, under pathological conditions, the immune system’s unbalanced activation could lead to tissue damage [75]. Antigen-presenting cells (APCs), such as macrophages and dendritic cells, are the ones responsible for sensing danger and initiating inflammatory processes, upon activation of PRRs by pathogen-associated molecular patterns (PAMPs) or damage-associated molecular patterns (DAMPs), and through antigen presentation to T cells. Microglia, the main tissue-resident macrophages, and therefore the predominant APCs of the brain, are the most abundant population of innate immune cells in the CNS, and during CNS inflammation they further contribute to the inflammatory process.

The involvement of the immune system in PD has been hypothesised since it was considered a multisystemic disorder, and evidence has grown significantly to support the hypothesis that (neuro)inflammation initiates and/or drives the progression of PD and that it occurs long before symptomatology becomes apparent. In 1988, McGeer and colleagues showed for the first time the presence of HLA-DR^+^ (a component of the human MHCII) reactive microglia in the post-mortem SNpc of PD patients [76]. Similarly, another group reported increased levels of activated microglia around dopaminergic neurons in MPTP-induced parkinsonism in post-mortem human brains [77]. Additionally, several groups demonstrated the existence of a pro-inflammatory phenotype or innate immunity activation of microglia through histological studies [78,79,80,81,82,83] and positron emission tomography (PET) imaging studies [84,85,86,87,88]. Extracellular α-synuclein can directly activate microglia early on, as shown in transgenic mice overexpressing wild-type or mutated α-synuclein [89,90]. Additional studies demonstrated that α-synuclein directly promotes microglia activation [91,92,93,94], inducing production and release of pro-inflammatory cytokines [91,92] and increasing expression of antioxidant enzymes [92]. Moreover, in addition to microglial activation, intranigral injection of α-synuclein also resulted in the upregulation of mRNA expression of major pro-inflammatory cytokines (IL-1β, IL-6, TNF-α) and the expression of endothelial markers of inflammation [94]. Furthermore, mitochondrial toxins, such as MPTP, 6-OHDA, and rotenone, have the ability to initiate an immune reaction in the striatum and SNpc, showing that specific damage to the ETC can trigger microglial activation and neuroinflammation [95,96,97,98]. Rotenone and 6-OHDA were depicted to activate microglia in mice [99], human microglial cell lines [100], and in rats [101,102,103].

Innate and adaptive immune responses are indeed activated in the CNS of PD patients and are known to cause upregulation of inflammatory responses. There is considerable evidence that inflammatory cytokines are secreted in the brain of PD subjects, as shown by increased levels of TNF-α in both the caudate and putamen [104]. Moreover, several studies performed on the cerebrospinal fluid (CSF) of PD patients revealed higher levels of IL-1β [105,106,107,108], IL-6 [105,106,107,109], and TNF-α [104,108,109]. Furthermore, higher levels of inflammatory cytokines are also found in the blood (the major pathway for immune cell circulation throughout the body) of PD patients, including IL-1β [110,111,112,113,114], IL-6 [110,112,115], TNF-α [112,116,117,118], and IFN-γ [117,118,119]. Together, these results validate the hypothesis that PD is indeed a multisystemic disorder.

As previously referred, α-synuclein pathology has also been found in the gut and in the vagus nerve. Chronic constipation, which is linked to peripheral inflammation [65], is one of the most frequent non-motor symptoms of PD. Enteric inflammation occurs in PD, and it has been demonstrated by increased levels of inflammation markers in the gastrointestinal tract, such as glial fibrillary acidic protein (GFAP) and pro-inflammatory cytokines, including IL-1β, IL-6, TNF-α, and IFN-γ, further reinforcing the role of peripheral inflammation in the initiation and/or progression of PD [120]. Remarkably, higher cytokine levels in the gut occur earlier during disease progression and decrease over time, possibly suggesting that gut inflammation could happen early during PD pathogenesis [121]. The occurrence of gastrointestinal dysfunction and gut inflammation in PD patients led to the investigation of variations in the gut microbiome of these individuals [121]. In fact, several studies have reported alterations in the gut microbiome of PD patients from multiple geographical populations [117,122,123,124,125,126,127]. For instance, *Roseburia* spp. and *Faecalibacterium* have been found to be decreased in PD stool. *Roseburia* spp. has been shown to have an anti-inflammatory role through (1) enhancement of intestinal barrier health [128], (2) downregulation of pro-inflammatory cytokines (e.g., IL-17) and differentiation of Treg cells [129], and (3) upregulation of anti-inflammatory cytokines (e.g., IL-10, TGFβ) [130]. Similarly, *Faecalibacterium* has been shown to reduce inflammation [131,132,133,134]. Together, these alterations indicate that the PD gut is more susceptible to inflammation. In general, the composition of gut microbiota is evidently altered in PD, and these findings are consistent with the hypothesis that gut dysbiosis is linked to an inflammatory environment that may contribute to the initiation and/or progression of PD pathology (Figure 2). The data presented suggest the crucial role of the immune system in both the pathogenesis and progression of PD, as several observations of aberrant innate and adaptive immune responses in the CNS, blood, and gut of PD patients link the immune system to PD risk.

## 4. The Interplay between Mitochondria and Innate Immunity

Upon microbial infection, the innate immune system is triggered through the recognition of PAMPs by PRRs expressed on the cells’ surface. Invading microorganisms and infected cells are then removed following an orchestrated pro-inflammatory immune response [65]. Nevertheless, PRRs have the ability to initiate innate immune responses independently of infection—a process known as sterile inflammation—through recognition of DAMPs, which are endogenous signals released upon injury or stress [135]. Mitochondria have been recognised as an important source of DAMPs; the reason behind this is the fact that mitochondria share a common origin with bacteria, and both display some similarities: circular DNA with CpG motifs, double-membrane structure with an abundance of cardiolipin in the inner membrane, secretion of N-formylated proteins, and reproduction by binary fission [135]. Upon injury or stress, the mitochondrial release of DAMPs activates the innate immune system, similarly to bacterial PAMPs, triggering sterile inflammation that mimics the response to infection [65,135].

Numerous studies have identified a critical role for mitochondria in regulating and activating the NLRP3 inflammasome [136], an intracellular immune sentinel activated upon changes in cellular homeostasis, which activates pro-inflammatory cytokines (IL-1β and IL-18) to trigger pyroptotic cell death [137,138]. Several studies demonstrate that mitochondrial dysfunction through ROS production is required for NLRP3 inflammasome activation [139,140]. Additional data have also revealed that mtDNA translocated to the cytosol can directly activate the NLRP3 inflammasome by binding to it [140,141]. Moreover, cardiolipin, a phospholipid found solely in inner mitochondrial and bacterial membranes [142,143], can also bind to and activate the NLRP3 inflammasome [144]. Mitochondria, along with mitochondrial DAMPs (mtDNA, ROS, and cardiolipin), are crucial for the activation and regulation of the NLRP3 inflammasome (Figure 3), which in turn incorporates mitochondrial dysfunction in a pro-inflammatory signalling response, thus elucidating the association of mitochondrial damage with inflammation [135]. Remarkably, exposure of neuronal and microglial cell lines to mitochondrial lysates led to an increase in inflammation markers, with mtDNA being proposed as the candidate DAMP, causing the inflammatory alterations detected [145]. Moreover, injecting mice with isolated mitochondria into the brain also resulted in an increase in inflammation markers, namely increased TNF-α, NF-κB phosphorylation, GFAP protein, and decreased *Trem2* mRNA [146]. 

The cause of the neuronal loss observed in PD brains is still poorly understood. Accumulating evidence leads to the hypothesis that the activation of innate immunity in dopaminergic neurons, through the exposure to DAMPs originating from mitochondrial dysfunction, bacteria, or even their metabolites targeting the mitochondria, could promote low-grade inflammation [135]. The mitochondrial network was found to be highly fragmented in cellular and animal models of PD [147], which is a pre-requisite for the selective degradation by mitophagy [147,148]. It was demonstrated that cardiolipin exposure, as a result of mitochondrial fission, is an important intervenient in the disposal of dysfunctional mitochondria [149]. Because cardiolipin is only present in mitochondrial or bacterial membranes, it has been considered a mitochondrial-derived DAMP that is detected by the NLRP3 [150]. Hence, cardiolipin exposure increases due to mitochondrial dysfunction, leading to the activation of neuronal innate immunity.

## 5. miRNA Involvement in Parkinson’s Disease

### 5.1. Human and Animal Studies

Cells, either within the same tissue or in different tissues/organs, can communicate across a long distance by sending information from one cell to another to coordinate their behaviours in order to grow, develop and survive [151]. Recent studies proposed that miRNAs contribute to cell-to-cell communication, by being secreted and transported to other cells via circulation to affect recipient cells [151]. This hypothesis has been validated by the detection of extracellular/circulating miRNAs in a multitude of biological fluids, such as blood, CSF, saliva, breast milk, urine, and others [152]. miRNAs can be found circulating in vesicles (exosomes, microvesicles, apoptotic bodies) or associated with proteins (AGO2), and, contrary to cellular RNA, extracellular miRNAs present high stability, providing a desirable characteristic for long-distance cellular communication [153]. It is now accepted that extracellular/circulating miRNAs can be used as biomarkers and as a therapeutic approach for a wide range of diseases but are also important in cell-to-cell communication.

In the last few years, miRNA dysregulation has been implicated in several neurodegenerative diseases, including PD, where it contributes to neurodegeneration and disease progression [153]. 

The importance of miRNAs for CNS integrity has been demonstrated by inducing a selective depletion of Dicer in midbrain dopaminergic neurons in mice, which impairs miRNA biogenesis and results in neurodegeneration and locomotor symptoms mimicking PD [154]. Furthermore, a multitude of screening studies have reported differentially expressed miRNAs in the brain [155,156,157,158,159,160], CSF [161,162,163,164], and blood [165,166,167,168,169,170,171,172,173,174] of PD individuals. In PD, some miRNAs have been associated with mitochondrial dysfunction, neuroinflammation, and dopaminergic neuron demise, thereby worsening disease pathogenesis. Moreover, several studies have demonstrated that specific miRNAs regulate PD-related genes, such as *SNCA*, *PRKN*, *DJ-1,* and *LRRK2*, modulating their functions in different cellular and animal PD models [175]. It has been observed that overexpression of miR-494 significantly decreased the levels of DJ-1 both in vitro, in 3T3-L1 and Neuro-2a cell lines, rendering cells more susceptible to oxidative stress, and in vivo, in a MPTP mouse model, exacerbating MPTP-induced neurodegeneration [176]. On the other hand, miR-7 was shown to exert a protective role by repressing expression of α-synuclein and accelerating the clearance of α-synuclein and its aggregates through autophagy [177]. The reduced levels of miR-7 in the SNpc of a MPTP PD mouse model correlated to nigrostriatal neurodegeneration and α-synuclein upregulation [178]. A protective role for miR-153 in PD was also hypothesised, since its overexpression decreased MPP^+^-induced neurotoxicity in murine cortical neurons [179]. Kabaria and co-workers observed that miR-34b and miR-34c bind to *SNCA* mRNA and reduce α-synuclein expression. In contrast, downregulation of these miRNAs led to increased α-synuclein levels and formation of α-synuclein aggregates. The same group also detected a polymorphism in the binding site of miR-34b, interfering with its binding and consequently leading to α-synuclein overexpression [180]. Both miR-34b and miR-34c have been found to be downregulated in PD brains [160]. In vitro studies in differentiated SH-SY5Y cells concluded that depletion of miR-34 b/c resulted in cell death associated with mitochondrial dysfunction and oxidative stress [160]. *LRRK2* gain-of-function mutations have been implicated in both genetic and sporadic PD [181,182]. Accordingly, LRRK2 levels were found to be increased in PD patients’ brains compared with healthy controls; however, no significant differences in LRRK2 transcripts were detected between both groups, suggesting a post-transcriptional modification of LRRK2 protein expression. In order to understand the reason behind this, Cho and colleagues proceeded to analyse the 3’UTR of *LRRK2* and found a binding site for miR-205, a miRNA found to be downregulated in the brain of PD patients. Further studies confirmed that miR-205 upregulation in cell lines and primary neuron cultures resulted in the downregulation of LRRK2, possibly having a protective effect in the brains of PD patients [156]. Additionally, mutated LRRK2 seems to be able to negatively regulate miRNA-mediated translational repression in *Drosophila melanogaster* brains. Briefly, mutated LRRK2 inhibits the expression of two miRNAs (let-7 and miR-184) known to target E2F transcription factor 1 and dimerization partner transcription factor, leading to defective cell division and neuronal death. These results indicate that mutated LRRK2 might also play a key role in PD pathogenesis by modulating the miRNA pathway [183]. On the other hand, miRNAs can directly or indirectly modulate the expression of PD-related genes. In fact, a recent study found that miR-421 targets Pink1. Mice treated with MPTP and SH-SY5Y cells treated with MPP^+^ were shown to overexpress miR-421. Downregulation of miR-421 attenuated neurodegeneration in MPTP-treated mice and promoted mitophagy in MPP^+^-treated SH-SY5Y cells, highlighting the role of miR-421 in regulating mitophagy via the Pink1/Parkin pathway [184]. Moreover, Zeng and colleagues highlighted the protective role of miR-135b, since it inhibits pyroptosis by targeting FoxO1 in MPP^+^-treated SH-SY5Y and PC-12 cells [185]. A report from 2018 showed that miR-494-3p negatively regulates sirtuin 3 (SIRT3) expression in both MPP^+^-treated SH-SY5Y cells and in a MPTP-induced PD mouse model, worsening motor impairment of these mice [186]. Another study revealed that miR-486-3p targets sirtuin 2 (SIRT2) and reduces its expression levels. However, the authors reported a PD risk-conferring polymorphism in the *SIRT2* gene and showed that a single nucleotide polymorphism (SNP) in this gene alters the binding efficiency of miR-486-3p to *SIRT2*, thereby increasing the expression level of SIRT2, which could increase α-synuclein aggregation and toxicity [187]. Furthermore, miRNAs have been shown to target several genes involved in neuroinflammation, a major hallmark of PD. Specifically, miR-155 was demonstrated to have an important role in the regulation of inflammation. The mentioned miRNA was found to be upregulated in a PD mouse model overexpressing α-synuclein. Deficiency of miR-155 prevented the increase in MHCII and the death of dopaminergic neurons triggered by α-synuclein overexpression. Oppositely, upregulation of miR-155 restored the inflammatory response to α-synuclein fibrils [188]. miR-7, in addition to having a role in the regulation of α-synuclein as previously described, emerged in the context of neuroinflammation as directly targeting the NLRP3 inflammasome gene. In fact, a study from 2016 determined that transfection of BV2 cells with miR-7 inhibited microglial NLRP3 inflammasome activation, while anti-miR-7 had the opposite effect, aggravating inflammasome activation in vitro. The same group reported that injecting miR-7 mimics directly into the striatum of a MPTP mouse model of PD suppressed NLRP3 inflammasome activation and ameliorated dopaminergic neuronal death [189]. Studies demonstrated the regulation of TNF-α levels by miR-7116-5p in a microglial cell culture model, where MPP^+^ potentiated TNF-α production by downregulating miR-7116-5p. Consistently, overexpressing miR-7116-5p in the microglia of an MPTP mouse model prevented the overproduction of TNF-α and the activation of glia, further reducing the loss of dopaminergic neurons [190]. Although research on this topic is still in its preliminary stages, and future studies are needed to better understand the role of miRNAs in neuroinflammation, these findings might help identify new therapeutic targets to downregulate microglial activation and potentially diminish dopaminergic neuron death in PD. Importantly, other non-coding RNAs may be involved in PD development progression. As an example, circular RNAs can function as miRNA sponges and act as competitive endogenous RNA to deregulate mRNA by miRNA. For instance, CircSLC8A1, which is increased in the SN of PD patients, regulates miR-128 function and/or activity affecting oxidative stress [191]. In addition, a recent study showed that transfer RNA fragment profiles, which were hypothesised to function as microRNAs, revealed disease-specific patterns in the CSF and blood of PD patients [192]. 

### 5.2. Gut Microbiota and microRNAs

Given the important involvement of gut dysbiosis and inflammation in PD and the potential of miRNAs to serve as diagnostic biomarkers, Kurz and colleagues [193] investigated the expression of miRNAs in routine colonic biopsies from PD patients and detected several differentially expressed miRNAs; amongst all, miR-486–5p upregulation showed the highest specificity for PD and correlated with age and disease severity in PD. In a follow-up analysis, 301 target genes of miR-486–5p were identified, as well as the biological processes affected by the mentioned miRNA, with brain development and post-synapse organization processes having the strongest functional association with the miR-486-5p target gene network [193]. 

The gut microbiota includes several microorganisms such as bacteria, viruses, protozoans, and fungi. Gut microbiota has a crucial role in the maintenance of gut homeostasis and integrity, since it interacts with the intestinal epithelial barrier (IEB) and intestinal epithelial cells [194]. Moreover, gut microbiota regulates intestinal epithelium growth, differentiation, and permeability [194]. Therefore, it comes as no surprise that alterations in gut microbiota composition can disrupt the IEB and lead to gut dysbiosis, with consequent immune and inflammatory response activation. Specific bacterial products (such as short-chain fatty acids (SCFAs), vitamins, or neurotransmitters) can interfere with the regulation of CNS immune and inflammatory processes, including microglial activity, by infiltrating into the bloodstream and traveling to the brain [195]. Additionally, they have the ability to directly activate circulating immune cells, which then travel to the CNS and regulate brain physiology [195]. Importantly, the gut–brain axis depends on a tightly regulated interplay between immunity and gut microbiota (Figure 2). As previously stated, it has been shown that gut microbiota is significantly altered in PD patients, compared with healthy controls; it is becoming evident that it might play an important role in the pathogenesis of PD. In addition, recent evidence indicates that miRNAs are involved in PD pathophysiology, suggesting a novel disease-associated mechanism that is now beginning to be explored and positing miRNAs as potential biomarkers for PD screening. Interestingly, miRNAs were found in human faecal samples, mainly derived from intestinal epithelial cells. Furthermore, it was recently demonstrated that secreted miRNAs are able to enter bacteria and regulate bacterial gene transcripts and affect their growth [196], giving rise to the possibility of the host’s miRNAs’ affecting and shaping their own gut microbiome in PD (Figure 4). This hypothesis was proposed after *Fusobacterium nucleatum* cultured with human miR-515-5p (a miRNA present in human faeces) showed an increased ratio of 16S rRNA/23S rRNA transcripts and altered growth [196]. Moreover, selective deletion of Dicer in mice promoted gut microbiota imbalance and exacerbation of dextran sulphate sodium (DSS)-induced colitis. These effects were reversed by faecal miRNA transplantation from wild-type littermates [196]. These studies further corroborate the hypothesis that miRNAs regulate gut microbiota and enhance the role of faecal miRNAs in influencing gut microbiota and preserving intestinal homeostasis. In order to better understand the potential interactions between miRNAs and the gut metagenome, Hewel and co-workers [197] performed an in silico target screen for binding sites of PD-associated miRNAs on human gut metagenome sequences, from which resulted a massive number of interactions. They found numerous miRNAs that may be key regulators in bacterial pathways relevant to PD, including the bacterial secretion system and lipopolysaccharide (LPS) biosynthesis [197]. On the other hand, gut microbiota, primarily through gut microbiota metabolites, may modulate human gene expression by affecting the host’s miRNA expression (Figure 4). In fact, Peck et al. demonstrated that the microbiota modulates miRNA expression in intestinal epithelium cells, which may alter intestinal homeostasis [198]. Different miRNA expression profiles were observed in the colon and ileum of germ-free mice colonised with gut microbiota from pathogen-free mice, when compared with germ-free littermates [199]. The analysis of faecal miRNA expression patterns also revealed significant differences between conventional mice and germ-free mice [200]. Additionally, it has also been demonstrated that depletion of gut microbiota using antibiotics induces alterations in faecal miRNA expression profiles in vivo [201]. A report from 2014 revealed that adherent-invasive *E. coli*, a pathogen highly prevalent in Crohn’s disease, upregulates miRNAs (miR-30c and miR-130a) that target genes involved in the autophagy response (ATG5 and ATG16L1) in mouse enterocytes, which may facilitate adherent-invasive *E. coli* replication and aggravation of intestinal inflammation [202]. Downregulation of miR-144 induced by *Lactobacillus casei* was reported to enhance intestinal barrier integrity through upregulation of occludin and zonula occludens 1 levels in intestinal epithelial cells, boosting intestinal barrier function and homeostasis [203]. Similarly, several probiotics have shown their ability to modulate miRNAs in intestinal epithelial cells and immune cells, altering intestinal barrier function and intestinal immune regulation [204,205,206]. A link between gut microbiota and brain miRNA expression has been established, since abnormal brain miRNA expression profiles have been described in the amygdala and prefrontal cortex of germ-free mice and mice treated with antibiotics to induce microbiota depletion [207]. Moreover, *Bacteroides fragilis* LPS, a microbial endotoxin, was shown to induce several miRNAs responsible for targeting genes that regulate synaptic architecture, amyloidogenesis, and brain inflammatory signalling [208]. These studies shed light on bidirectional communication between gut microbiota and miRNAs. Since both gut dysbiosis and differently expressed miRNAs have been reported in PD, the bidirectional interaction of gut microbiota–miRNAs might be involved in the pathophysiology of PD. We can hypothesize that microbial dysbiosis can potentiate the dysregulation of certain miRNAs, which can then target mitochondria and inflammatory pathways in the gut and later on in the brain, triggering PD pathological hallmarks (Figure 5).

## 6. Conclusions and Future Perspectives

The complexity and heterogeneity of PD have contributed to the challenging nature of PD diagnosis and treatment. The currently used diagnostic methods for PD are based on clinical symptoms that only emerge when the disease has already progressed to a stage of elevated neuronal loss. Therefore, there is a growing need to identify molecular biomarkers that allow an early and precise diagnosis. The possibility of using miRNAs as potential biomarkers for the diagnosis of PD during the prodromal phase gained relevance with the emergence of the hypothesis that miRNAs are involved in the pathogenesis and pathophysiology of PD. Several promising candidate miRNAs have surfaced from the numerous screenings performed so far on CSF, blood, and brain tissue of PD individuals or models. However, obtaining these samples is either invasive (CSF) or only possible post-mortem (brain tissue), making blood samples the ideal source of biomarkers, because they are easily extracted and analysed and facilitate patient monitoring over time. Moreover, considering the involvement of the gut in the aetiology and pathophysiology of PD, studying dysregulated miRNAs in the gut or even in the faecal material of PD patients might lead to the surfacing of an important and useful strategy for the early detection of gut-first PD and for the development of therapies able to slow or halt disease progression before it reaches the brain.

However, although microRNAs show potential to be reliable biomarkers, most of the candidate miRNAs are not organ-specific, and their profile is highly dependent on uncontrollable factors such as individual genetics and ethnicity, as reported in a study from 2020, where differences were observed between different ethnic cohorts of PD patients (American, Asian, and European) [209].

The potential of using miRNA-based therapies for the treatment of PD has also gained importance over the years. The scientific community has been developing strategies for miRNA modulation for many years, by using miRNA-mimics and antago-miRs to upregulate or downregulate miRNA levels, respectively. Nonetheless, the path to develop miRNA-based therapies has been paved with numerous challenges, including the lack of target specificity of miRNAs and their delivery to specific sites [210]. Unravelling relevant targets of miRNAs in PD models, followed by human validation of the results, may accelerate finding new biomarkers for an early diagnosis and novel therapeutic strategies for PD. To find a promising gut miRNA that can be a PD biomarker candidate would allow for therapeutic intervention in the prodromal phase of these patients, stopping PD progression and development. Hence, further studies of the microbiome–miRNA axis are a must in PD research.

## Figures and Tables

**Figure 1 biomedicines-11-01349-f001:**
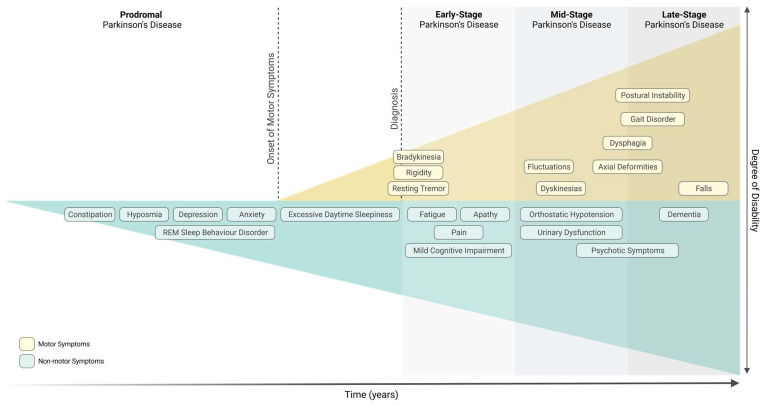
Motor and non-motor symptoms associated with Parkinson’s disease. Diagnosis of Parkinson’s disease occurs with the onset of motor symptoms (early-stage Parkinson’s disease) but can be preceded by a prodromal phase of several years, which is characterised by specific non-motor symptoms (prodromal Parkinson’s disease). Created with Biorender.

**Figure 2 biomedicines-11-01349-f002:**
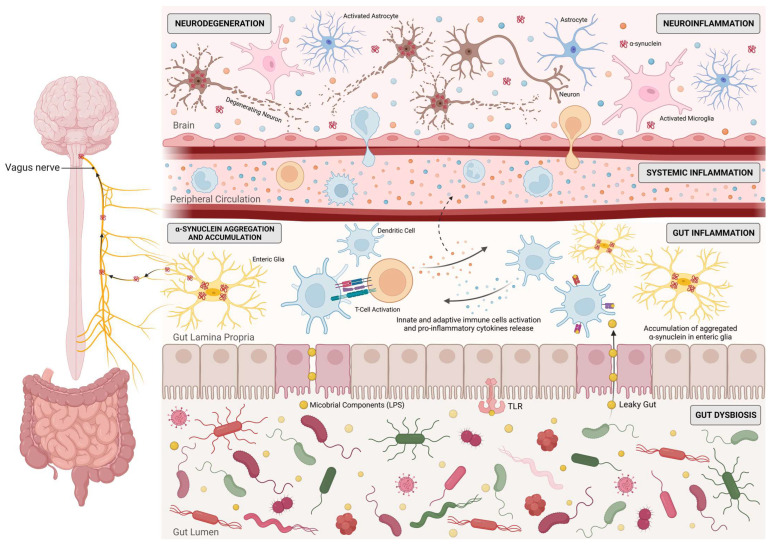
Dysregulation of the microbiota–gut–brain axis in the pathogenesis of Parkinson’s disease. Gut dysbiosis increases intestinal barrier permeability and leakage of bacterial products from the lumen to the lamina propria, activating inflammatory responses and leading to gut inflammation. These inflammatory processes induce misfolding of α-synuclein, which results in its aberrant aggregation and accumulation in the gut and enteric neurons. α-synuclein aggregates are then transported from the gut to the brain through the vagus nerve, eventually triggering an inflammatory response of the microglia. Alternatively, pro-inflammatory cytokines released during gut inflammation might infiltrate the bloodstream, leading to a systemic inflammation that further increases the permeability of the blood–brain barrier and allows the infiltration of immune cells into the brain. These processes result in neuroinflammation and consequent neurodegeneration. Created with Biorender.

**Figure 3 biomedicines-11-01349-f003:**
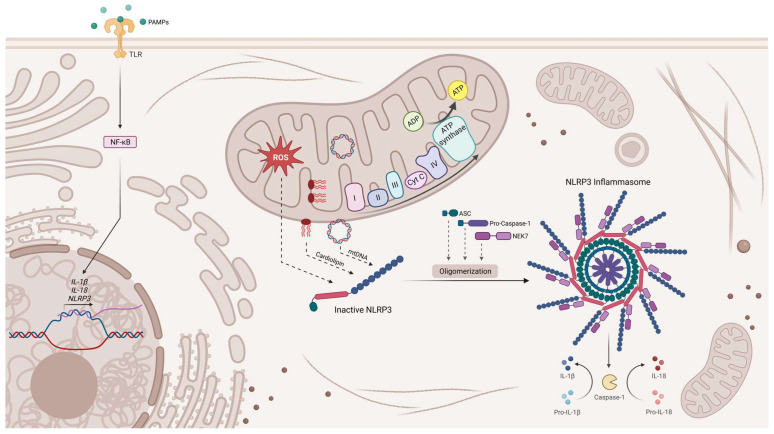
NLRP3 inflammasome activation. The activation of the NLRP3 inflammasome is triggered by a number of pathogen-associated molecular patterns (PAMPs) or damage-associated molecular patterns (DAMPs). PAMPs bind to Toll-like receptors (TLR) present on the cell membrane and upregulate the transcription of NLRP3 inflammasome components. DAMPs, such as ROS, mtDNA, or the externalization of cardiolipin to the outer mitochondrial membrane, also activate NLRP3. Assembly of the inflammasome activates caspase 1, which in turn cleaves pro-IL-1β and pro-IL-18 into IL-1β and IL-18, respectively. Created with Biorender.

**Figure 4 biomedicines-11-01349-f004:**
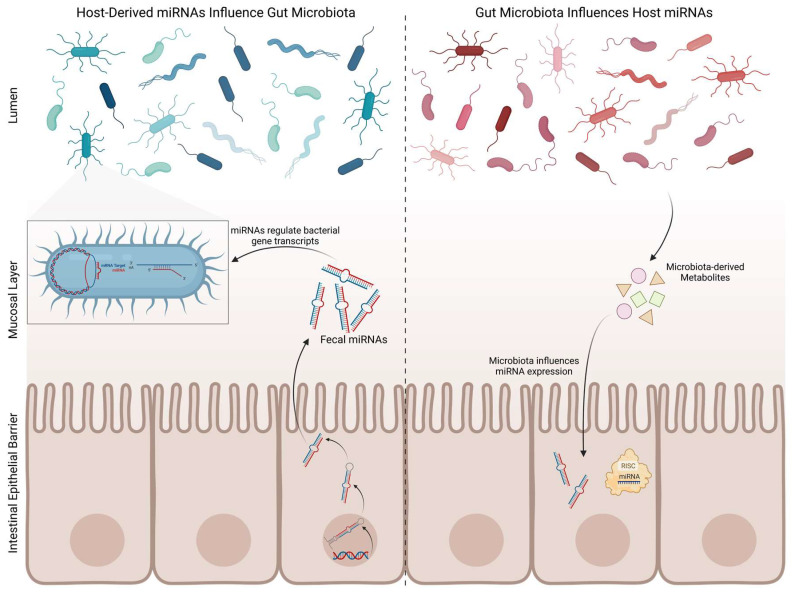
Reciprocal regulation of miRNAs and gut microbiota. Host’s intestinal epithelial cells release miRNAs that have the ability to regulate bacterial gene transcripts, affecting bacteria growth and replication. On the other hand, microbiota regulates host’s miRNA expression. Created with Biorender.

**Figure 5 biomedicines-11-01349-f005:**
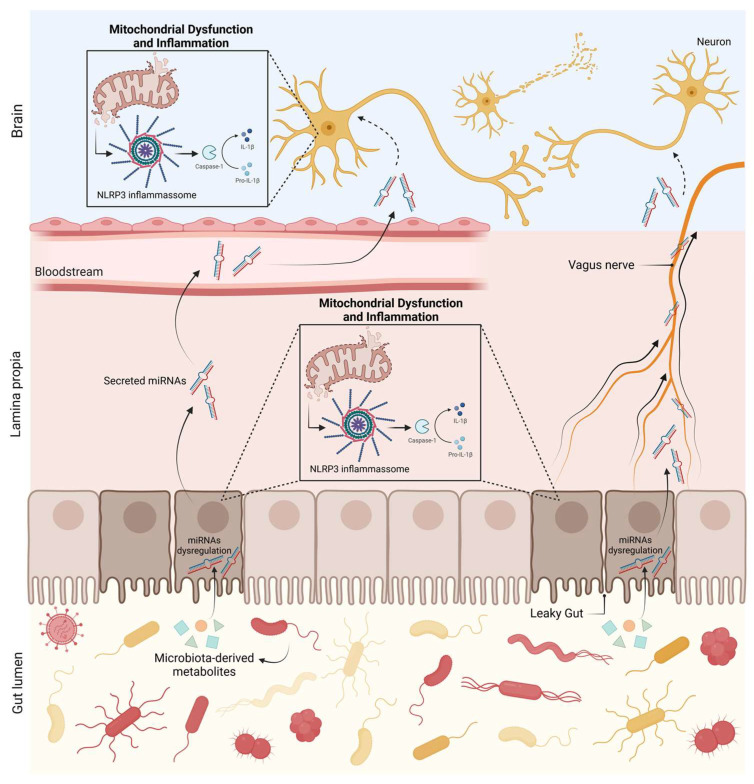
miRNA–microbiota axis in PD. Microbial dysbiosis can potentiate the dysregulation of certain miRNAs, which can then target mitochondria and inflammatory pathways in the gut, leading to a proinflammatory response and to the loss of intestinal barrier integrity. This will allow miRNAs to travel freely or within vesicles through the blood or through the vagus nerve and reach the brain. Within the brain, miRNAs can target mitochondria and activate neuronal innate immunity, ultimately leading to PD hallmarks. Created with Biorender.

## Data Availability

Not applicable.

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
