# Peer review of "The Impact of microRNAs on Mitochondrial Function and Immunity: Relevance to Parkinson’s Disease"

_biomedicines, 2023, doi:10.3390/biomedicines11051349_

Round 1
Reviewer 1 Report
The author summarize the impact of miRNA in mitochondria function and immunity, which is related to PD pathogenesis. This topic is interesting.
However, there are some serious drawbacks.
The authors only discussed the roles of microRNA, mitochondria and gut microbiota in PD pathogenesis respectively. The contents on the roles of miRNA in mitochondria and the roles of miRNA in immunity, related to PD pathogenesis is very less.
There are some recent published review papers on miRNA and mitochondria and immunity (MicroRNAs Dysregulation and Mitochondrial Dysfunction in Neurodegenerative Diseases. Int. J. Mol. Sci. 2020, 21, 5986. MicroRNAs Play a Role in Parkinson’s Disease by Regulating Microglia Function: From Pathogenetic Involvement to Therapeutic Potential. Front Mol Neurosci. 2021; 14: 744942. MicroRNA Dysregulation in Parkinson’s Disease: A Narrative Review). In these papers, more detailed discussions have been provided, especially the impacts of miRNA on mitochondria in PD. The content in this review is not as good as what have been published previously.
On the other hand, the mitophagy deregulation can also be a pathogenic factor in PD. However, the authors have not mentioned it at all.
Therefore the review is not up to the request to be published in the current journal.
Author Response
Reviewer 1:
The author summarize the impact of miRNA in mitochondria function and immunity, which is related to PD pathogenesis. This topic is interesting.
However, there are some serious drawbacks.
R: We thank the reviewer for the input and suggestions.
The authors only discussed the roles of microRNA, mitochondria and gut microbiota in PD pathogenesis respectively. The contents on the roles of miRNA in mitochondria and the roles of miRNA in immunity, related to PD pathogenesis is very less.
There are some recent published review papers on miRNA and mitochondria and immunity (MicroRNAs Dysregulation and Mitochondrial Dysfunction in Neurodegenerative Diseases. Int. J. Mol. Sci. 2020, 21, 5986. MicroRNAs Play a Role in Parkinson’s Disease by Regulating Microglia Function: From Pathogenetic Involvement to Therapeutic Potential. Front Mol Neurosci. 2021; 14: 744942. MicroRNA Dysregulation in Parkinson’s Disease: A Narrative Review). In these papers, more detailed discussions have been provided, especially the impacts of miRNA on mitochondria in PD. The content in this review is not as good as what have been published previously.
R: We understand the reviewer concern, however our review is intended to provide an overview of mitochondrial dysfunction and inflammation deregulation, highlighting the crosstalk between the two in Parkinson’s disease and, depict the involvement of miRNAs. Moreover, to our knowledge this is the first report emphasizing the possible importance of miRNA and microbiota crosstalk in neurodegenerative disorders such as PD. None of the refereed reviews discusses mitochondria and inflammation cross-talk in PD or describes miRNA-microbiota axis potential role in PD.
Indeed, the review Front Neurosci., 2020 shows a more descriptive revision of miRNA deregulation involvement in PD which is not the aim of our review. The review Front Mol Neurosci., 2022 discusses the role of microglia dysfunction in the progression of PD and describes miRNAs involved mainly in this process, which is out of the scope of this revision. In Int. J. Mol. Sci., 2020 the authors provide a revision of miRNA deregulation only in mitochondrial dysfunction in several neurodegenerative diseases with no reference to the cross-talk between mitochondria and immunity.
On the other hand, the mitophagy deregulation can also be a pathogenic factor in PD. However, the authors have not mentioned it at all.
R: As suggested by reviewer 1 we discussed mitophagy involvement in PD. Please see Page 5, Line 210-218.
Therefore, the review is not up to the request to be published in the current journal.

Reviewer 2 Report
This is an interesting review, which tackles both well established and relatively new information on the links of Parkinsonism with immune system changes, inflammation and microbiota as those relate to microRNA alterations. These are all positive aspects and the text reads fluently. However, there are some issues that call for revision, as is listed below.
11. Male to female differences in the susceptibility to PD deserve to be described in a little deeper approach, especially since the failure of mitochondria, which is notably distinct in men and women living with PD seem to be most relevant to this disease.
22. Apart from microRNAs, other non-coding RNAs may also affect PD progression. For example, circular RNAs and their capacity to ‘sponge’ disease-affecting microRNAs need to be referred to (as for example in Hanan et al., 2020). Likewise, many of the recently re-discovered transfer RNA fragments (tRFs) may operate like microRNAs and present age-related changes in PD patients’ CSF (as in Paldor et al., 2022).
33. The susceptibility of diverse ethnic populations to PD is likely reflecting the incidence of specific mutations (e. g. in the LRRK2 gene) in those populations; however, the authors did not refer to the possibility of the PD-related microRNAs being differently presented in such distinct populations. This topic as well merits discussion.
44. The figures are all artistic and impressive, but they are less related to the molecular focus of this review. Adding a figure or two to highlight the proposed pathways can be helpful.
Author Response
Reviewer 2:
This is an interesting review, which tackles both well established and relatively new information on the links of Parkinsonism with immune system changes, inflammation and microbiota as those relate to microRNA alterations. These are all positive aspects and the text reads fluently. However, there are some issues that call for revision, as is listed below.
R: We thank the reviewer for the comments and suggestions.
- Male to female differences in the susceptibility to PD deserve to be described in a little deeper approach, especially since the failure of mitochondria, which is notably distinct in men and women living with PD seem to be most relevant to this disease.
R: As suggested we added information regarding male to female differences in PD susceptibility highlighting mitochondria involvement. Please see Page 5, Line 230-235.
- Apart from microRNAs, other non-coding RNAs may also affect PD progression. For example, circular RNAs and their capacity to ‘sponge’ disease-affecting microRNAs need to be referred to (as for example in Hanan et al., 2020). Likewise, many of the recently re-discovered transfer RNA fragments (tRFs) may operate like microRNAs and present age-related changes in PD patients’ CSF (as in Paldor et al., 2022).
R: As suggested we added examples of other non-coding RNAs that can affect PD progression and development. Please see Page 12, Line 507-514.
- The susceptibility of diverse ethnic populations to PD is likely reflecting the incidence of specific mutations (e. g. in the LRRK2 gene) in those populations; however, the authors did not refer to the possibility of the PD-related microRNAs being differently presented in such distinct populations. This topic as well merits discussion.
R: As suggested we discussed this topic. Please see Page 16, Line 625-629.
- The figures are all artistic and impressive, but they are less related to the molecular focus of this review. Adding a figure or two to highlight the proposed pathways can be helpful.
R: As suggested we added a Figure highlighting the proposed pathways. Please see Figure 5.

Reviewer 3 Report
The authors reported an interesting review on the role of microRNAs in mitochondrial function and immunity in Parkinson’s Disease. I have some comments to the authors:
- In the fourth paragraph of the introduction, describing the possible risk and protective factors of PD, it would be helpful to include the possible role of these factors in the development of the clinical features. This relationship should be highlighted with a brief sentence in the text, here the related reference:
Belvisi D, et al. Relationship between risk and protective factors and clinical features of Parkinson's disease. Parkinsonism Relat Disord. 2022
- In the last paragraph of the introduction, the authors should state how the review have been performed (e.g. the methodological research of the literature) and the type of the review (e.g. narrative review).
- The authors should shorten the manuscript, it is very dense.
- Please provide a clear-cut “take home message” in the conclusion.
Author Response
Reviewer 3:
The authors reported an interesting review on the role of microRNAs in mitochondrial function and immunity in Parkinson’s Disease. I have some comments to the authors:
R: We thank the reviewer for the comments and suggestions.
- In the fourth paragraph of the introduction, describing the possible risk and protective factors of PD, it would be helpful to include the possible role of these factors in the development of the clinical features. This relationship should be highlighted with a brief sentence in the text, here the related reference: Belvisi D, et al. Relationship between risk and protective factors and clinical features of Parkinson's disease. Parkinsonism Relat Disord. 2022
R: We followed the reviewer suggestion and added a brief sentence in the text describing the possible risk and protective factors in the development of PD clinical features. Please see Page 2, Line 86-90.
- In the last paragraph of the introduction, the authors should state how the review have been performed (e.g. the methodological research of the literature) and the type of the review (e.g. narrative review).
R: We added this information. Please see Page 4, Line 135-144.
- The authors should shorten the manuscript, it is very dense.
R: Taking into account that we added all the reviewers suggestions and that we do believe that taking out some information would harm the relevance of this review we were not able to shorten the manuscript. We apologize for this.
- Please provide a clear-cut “take home message” in the conclusion.
R: We followed the reviewer suggestion and added a take home message in the conclusion. Please Page 16, Line 638-641.

Round 2
Reviewer 1 Report
I think that the revised manuscript has been improved significantly, so can be accepted for publication currently.